# Finite Element Analysis of Split Sleeve Cold Expansion Process on Multiple Hole Aluminum Alloy

**DOI:** 10.3390/ma16031109

**Published:** 2023-01-27

**Authors:** Yuan Lv, Meng’en Dong, Teng Zhang, Changkai Wang, Bo Hou, Changfan Li

**Affiliations:** 1College of Mechanical Engineering, Xi’an University of Science and Technology, Xi’an 710064, China; 2Aviation Engineering School, Air Force Engineering University, Xi’an 710038, China; 3Army Aviation Institute of PLA, Beijing 101100, China

**Keywords:** split sleeve cold expansion, compressive residual stress, 7075 aluminum alloy, finite element method, digital image correlation

## Abstract

Multiple cold expansion holes are widely used in connection areas of aircraft structures, in order to achieve uniform load transfer of the skin or connection parts. Split sleeve cold expansion (SSCE) is widely used to enhance the fatigue life of fastener holes by applying compressive residual stresses around the holes. In this study, the finite element method (FEM) was used to research the distribution and variation of residual stresses along the hole edges of 7075AA single-hole and multi-hole cold expansion (CE) specimens. Full-field strain measurements of single-hole and multi-hole specimens were performed using two-dimensional digital image correlation (DIC), and the residual stress and strain at the hole edge of the specimens measured by FEM and DIC were compared. FEM results shows that the maximum circumferential and radial residual stresses of three-hole specimens with three-hole spacing are increased by 5.37% and 31.53% compared with single-hole specimens. The maximum circumferential residual stress of three-hole specimens with four-hole spacing increases by 7.25% compared with a single hole, but the radial residual stress decreases by 12.98%. In addition, for three-hole specimens with hole spacing three times the hole diameter, the strengthening effect of SSCE in the order of middle hole, then left hole, and, finally, right hole is better than that of SSCE in the order of left to right hole. FEM and DIC full-field strain results are basically consistent.

## 1. Introduction

Bolts and rivets are used in the assembly of aircraft structures because of their versatility and high degree of interchangeability [1,2]. However, during aircraft service, the holes of rivets and bolts produce material discontinuities and severe stress concentrations [3]. Under fatigue loading, the hole becomes a weak link and the structure becomes prone to fatigue fracture and causes accidents in severe cases [4]. Cold expansion (CE) is an important measure that improves the fatigue life of aircraft structures [5]. By passing a conical mandrel through the fastening hole, CE creates a compression residual stress field around the hole, which enhances the fatigue resistance of the fastening hole [6,7,8]. However, this method limits the improvement of fatigue life due to surface damage caused by the direct contact between the mandrel and the workpiece hole surface [9,10].

The addition of split sleeves is generally considered a better method to protect the hole edge material from tearing and strong friction during mandrel and ball expansion [11,12,13]. Split sleeve cold expansion (SSCE) can not only avoid axial scratching of hole walls, but also has the advantages of good adaptability and high productivity, which lead to the wide application of SSCE in the aircraft assembly field [14,15]. In SSCE, the split sleeve is placed between the hole and the mandrel, the mandrel squeezes the sleeve which expands against the hole wall, and it produces circumferential compressive residual stress at the hole edge [16]. When the structure is loaded, the sleeve effectively reduces the maximum stress at the hole edge and improves the overall fatigue life of the fastener hole plate [17,18].

The finite element method (FEM) is an efficient method to simulate the SSCE process of split sleeves [19,20,21]. Dey et al. developed a three-dimensional finite element model of SSCE and studied variable material properties and expansion percentage parameters. Their study revealed that the yield strength of aluminum specimens and the percentage of CE of split sleeves improved the effect of annular residual stresses around the hole [22]. Lu et al. studied the effect of composite hole expansion treatment. A three-dimensional (3D) finite element model was developed for studying the split sleeve expansion process [23]. Zhang et al. established a three-dimensional finite element model to simulate the SSCE process considering non-uniform stress distribution of the split sleeve, and found that the compressive stress value is the largest in the middle layer, the smallest in the squeeze layer, and the value in the extrude layer is between the two [24]. Based on numerical simulation and experimental testing methods, Seifi et al. studied the effect of CE-induced residual stress of two adjacent holes on fatigue life and crack initiation period and analyzed the effect of expansion ratio and distance between adjacent holes on total life and crack growth life [25,26]. Kumar et al. conducted a 3D finite element simulation of continuous CE of two adjacent holes in an Al 7075-T651 plate. They compared the CE-induced predicted residual stress around the hole with experimental results [27]. At present, most published studies are about single-hole or two-hole cold expansion finite element analysis [27,28,29]. However, since a single hole or two holes cannot represent the effects of residual stress interference between multiple holes in actual aircraft service, there is an urgent need to conduct multi-row hole studies. Digital image correlation (DIC) is a measurement technique for photometric mechanical deformation. It analyzes pre- and post-deformation scattered images of the object surface by matching and tracking the motion of geometric points on the scattered images and determines the displacement and strain fields [18]. One of the most attractive features of DIC is its ability to measure strain under full-field conditions [30,31]. At present, scholars have used DIC to carry out research on CE-related issues [32,33,34]. However, the SSCE process of three holes has not been studied in depth, and the DIC method is rarely used to measure the residual strain at the hole edge of multi-hole samples during the SSCE process.

In this study, the SSCE of three holes in Al7075-T6 was studied by using ABAQUS. The purpose of this study is to analyze the compressive residual stress at the hole edge of multi-hole specimens during the SSCE process by an economical and efficient method. Three-dimensional elastic–plastic finite element simulations are carried out to obtain the stress and strain fields around the holes of specimens during SSCE. The simulation includes the influence of SSCE on residual stress under different hole spacings and sequences of three holes, and the influence of residual stress interference between adjacent holes of multi-hole specimens compared with single-hole samples. The FME process takes into account the interference of stress field between multiple holes under actual working conditions. To verify the accuracy of simulation results, the full-field full strain was measured by the DIC method, and the simulation results were compared with the experimental data. The results of single-hole and multi-hole SSCE were analyzed to evaluate the feasibility of multi-hole SSCE in actual working conditions.

## 2. Materials and Methods

### 2.1. Material and Test Piece Parameters

To reduce the influence of the residual stress of aluminum alloy plate matrix on the CE-induced residual stress of the hole, an Al7075-T6 plate is used without an aluminum cladding layer. The processing and manufacturing of the specimens were carried out by Southwest Aluminum (Group) Co., Ltd Xi’an, China. The material preparation process is as follows: (1) the ingot is heated to 400 °C for rolling; (2) solution treatment at 470 °C for 2 h and quenching in water at room temperature; (3) 2% pre-deformed; (4) aging treatment at 120 °C for 16 h. The chemical composition of Al7075-T6 is given in Table 1. The thickness and mechanical properties of the Al7075-T6 plate are given in Table 2. The test pieces used in this study are extracted from a cold-expanded aluminum alloy plate with a central hole and three holes at different hole spacings (three times and four times the hole diameter). The experimental work includes the investigation of the residual stress distribution law at the hole edge of single-hole and three-hole cold-expanded specimens and the effect of different hole spacings and expansion sequences on the stress interference between adjacent cold-expanded holes. Figure 1 shows the schematic diagram of the single-hole aluminum alloy specimen (without CE). The three-hole aluminum alloy specimens with different hole spacings (without CE) are shown in Figure 2, where (a) the hole spacing is 3 times the final hole diameter and (b) the hole spacing is 4 times the final hole diameter. The three-hole cold-expanded aluminum alloy specimens are shown in Figure 3. The thickness of aluminum alloy sheet specimens is 4 mm.

For the SSCE, the amount of extrusion is the main parameter. Two different expression methods were used one for the absolute extrusion amount $E_{a}$ and the other for the relative extrusion amount $E_{r}$. These can be expressed as:

Absolute extrusion amount:(1)$$E_{a}=d_{0}+2\delta -D_{0}$$

Relative extrusion amount:(2)$$E_{r}=\frac{E_{a}}{D_{0}}\times 100\%$$
where $d_{0}$ is the diameter of the working section of the extrusion mandrel, δ is the wall thickness of the split sleeve, and $D_{0}$ is the diameter of the initial hole.

In the CE process, the amount of extrusion of the split sleeve depends on the strength of the material to be extruded, the hole diameter, the distance between the holes (e/D, the ratio of the distance from the center of the hole to the smallest edge of the part to the aperture), hole spacing, and some other factors. For varied materials and different apertures, different extrusion amounts are required to precisely predict the fatigue life.

### 2.2. Strain Definition

Only in the case of limit Δl→dl→0 is the strain under tension and compression the same, which is
(3)$$d_{\varepsilon }=\frac{dl}{l}$$
(4)$$\varepsilon =\int_{l_{0}}^{l}\frac{dl}{l}=\ln\left(\frac{l}{l_{0}}\right)$$
where l is the current length; $l_{0}$ is the initial length; ε is the true strain or logarithmic strain.

The stress measure that is conjugated with the true strain is called the true stress and is defined as:(5)$$\sigma =\frac{F}{A}$$
where F is the force applied to the material; A is the current area.

When defining plastic data in ABAQUS, true stress and true strain must be used, while ABAQUS converts nominal stress and nominal strain from true values [35]. In order to establish the relationship between the true strain and nominal strain, the nominal strain is first expressed as:(6)$$\varepsilon_{nom}=\frac{l-l_{0}}{l_{0}}=\frac{l}{l_{0}}-\frac{l_{0}}{l_{0}}=\frac{l}{l_{0}}-1$$

The relationship between true strain and nominal strain can be obtained by adding 1 to the equation and removing the natural logarithm:(7)$$\varepsilon =\ln\left(1+\varepsilon_{nom}\right)$$

Considering the incompressibility of plastic deformation, the relationship between the real stress and nominal stress is:(8)$$l_{0}A_{0}=lA$$

Substituting the definition of A into the defining equation of true stress obtains:(9)$$\sigma =\frac{F}{A}=\frac{F}{A_{0}}\frac{l}{l_{0}}=\sigma_{nom}\left(\frac{l}{l_{0}}\right)$$
where $\frac{l}{l_{0}}$ can also be written as $1+\varepsilon_{nom}$, and the relationship between real stress, nominal stress, and nominal strain can be obtained by substituting it into the above equation:(10)$$\sigma =\sigma_{nom}\left(1+\varepsilon_{nom}\right)$$

In this study, when adding material properties to aluminum plates, mandrel, and split sleeve in ABAQUS, the stress and strain are both real stress and real strain.

### 2.3. Finite Element Model

The specimen, split sleeve, and mandrel are modeled as 3D deformable bodies, but rigid body constraints need to be created for the mandrel. The Young’s modulus of AL7075-T6 is 71.7 GPa, Poisson’s ratio is 0.33, and the true stress–strain curve is shown in Figure 4. Both the split sleeve and mandrel are made of ordinary steel with a Young’s modulus of 210 GPa and a Poisson’s ratio of 0.295.

The numerical models of specimen, mandrel, and split sleeve are developed in ABAQUS. All models are developed with hexahedral elements, 47,712 cells for the specimen with a central hole, 67,680 cells for a three-hole CE with a triple bore distance, 64,080 cells for a three-hole CE with a quadruple bore distance, 290 cells for the split sleeve, and 1700 cells for the mandrel. The smallest unit is the hole edge unit. The mandrel is considered a rigid body, while the plate and split sleeve are defined as deformable bodies. The contact pairs are defined by considering the contact pair of the plate and split sleeve, and the contact pair of the sleeve and mandrel. The friction coefficient between the sleeve and mandrel is 0.1.

The simulation analysis step adopts explicit dynamics. The time period of the single-hole CE process is set to 0.1 s, and the time period of the three-hole CE process is set to 0.3 s. The outer wall of split sleeve is in surface-to-surface contact with the hole wall. To simulate the split sleeve lubrication in a real SSCE process, the friction coefficient is set to 0. The mandrel is in surface-to-surface contact with the inner wall of the split sleeve, the friction coefficient is set to 0.1. By setting a displacement of 20 mm for a single mandrel, the simulation makes the mandrel working section extrude the split sleeve and pass completely. The three-hole SSCE process uses amplitude to control the extrusion sequence and time. According to the extrusion sequence, one mandrel is set to complete extrusion every 0.1 s, and three mandrels complete SSCE for three holes after 0.3 s.

In this study, the simulation focuses on the residual stresses in the perimeter direction of the hole. It is necessary to refine the mesh around the hole. To ensure mesh convergence, the circular area around the hole is divided separately, the large square area is further divided on the outside of the circle, and the square area where each hole is located is divided further into four parts. To reduce the computational effort, the seeds are distributed in a single-bias way to achieve dense seeds at the edge of the hole and sparse seeds away from the edge of the hole. The bias ratio is 10 and the number of elements on each hypotenuse is 20. Figure 5 shows the results of the mesh division around the hole of the central-hole specimen. Figure 6 shows the mesh quality around the three-hole specimen.

### 2.4. Test Equipment and Methods

The entire process of the hole SSCE test is recorded by DIC, a high-speed camera using phantom v2512, with a Nikon 50 mm camera lens. The high-speed camera resolution is 1 million. The camera and LED lighting equipment face the test specimen, the camera focus is adjusted at the start of the SSCE process, and a video is recorded. Before the experiment, the specimen is sprayed with scattered spots. Subsequently, the scattering analysis software is applied to process and analyze the von Mises strain of the scattering image on the specimen, and results of the hole edge strain under different hole spacings are compared, and CE order is followed by observing the change in hole edge strain field. The SSCE test is performed with Material Test System (MTS) 810 fatigue testing (manufactured by MTS, Minnesota, USA). A collet with a clamping range of 0~10 mm is selected, and the prepared specimen is clamped on the MTS fatigue tester for the CE test, as shown in Figure 7.

The test procedure adopted in this study follows the FTI-8101 standard and CE process specification [36]. Before and after the hole SSCE, the hole is checked with the FTI go/no-go gauge. The go/no-go gauge is shown in Figure 8. Before SSCE, the no-go gauge is inserted into each hole of the test piece. If the mandrel does not go through the hole, the test piece is considered qualified. At the end of the SSCE, the go gauge passes through the hole. The diameter of the initial split sleeve hole is 5.738. The equipment is shown in Figure 9. The hydraulic oil pump model is FT-E102, the hole extrusion gun model is LB-30 S/N 0818, the split sleeve model is CBS-6-3-N-16F, the mandrel model is CBM-6-3-N-1-4-V1, the thickness of the split sleeve is 0.152 mm, and the diameter of the working section of the mandrel is 5.664 mm.

The SSCE process is shown in Figure 10. First, the split sleeve is mounted on the mandrel, then the mandrel is passed through the hole and the pull gun is started. The mandrel is squeezed through the hole, and the SSCE action is completed. The SSCE procedure must be restarted if the split sleeve is axially displaced or damaged. The residual stress field generated by different SSCE sequences is used for the analysis of three-hole specimens.

## 3. Results and Discussion

### 3.1. Finite Element Simulation

The equivalent von Mises stress of the single-hole cold-expanded specimen is shown in Figure 11. The equivalent von Mises stresses of each hole along the *x*-axis of the three-hole cold-expanded specimen with a hole spacing of three times the hole diameter and four times the hole diameter are shown in Figure 12 and Figure 13, respectively. From the figures, it can be observed that stress decreases outward from the hole edge, and the stress distribution of the mandrel is inconsistent at different distances from the hole center. However, the stress distribution at the edge of the adjacent hole at the opening of the split sleeve is maximum, and it decreases outwards. This is because the opening of the split sleeve keeps opening outwards when the mandrel extrudes through the split sleeve which results in stress concentration. To analyze the residual stress distribution, the extruded specimen is divided into three parts: the entry plane, mid-thickness plane, and exit plane. In addition, it can be observed that the residual stresses extending through the depth of the three holes are significantly different. The stress at the extruding end is higher than that at the middle layer and extruding end. This shows that the deformation around the hole during the extrusion process is uneven. The reason for this unevenness is the opening of the split sleeve during the extrusion, which hinders the extrusion of the specimen at the opening of split sleeve, called a ridge [37]. In the actual extrusion test, the hole wall material is accumulated plastically at the casing slit. Therefore, the root of the convex ridge produces microcracks and residual tensile stress. Figure 14 shows the plastic accumulation phenomenon of the opening angle of the split sleeve at the hole edge of the specimen, forming a ridge.

The circumferential stress and radial residual stress distributions of the minimum cross-section (at the hole center of the single-hole specimen), after the core rod extrudes the hole, are shown in Figure 15. The circumferential stress distribution and radial stress distribution of the minimum cross-section, after the mandrel extrudes the three-hole specimens with hole spacing three times the diameter of the hole, are shown in Figure 16. It can be observed that the circumferential residual stresses between the holes of three-hole specimens with hole spacing three times the diameter of the hole interfere with each other, and the maximum stress occurs on the right side of the left hole, the left side of the central hole, the left side of the right hole, and the right side of the central hole. This shows that the residual stress produced by SSCE generates partial superposition when adjacent holes are extruded. The radial stress distribution direction is shown in Figure 15b, which is perpendicular to the direction of the principal load on the specimen. The circumferential stress and radial stress distributions of the minimum cross-section, after the mandrel extrudes the three-hole specimens with hole spacing four times the diameter of the hole, are shown in Figure 17. On the test piece, the circumferential residual stress interference of the three-hole specimen with spacing four times the diameter of the hole is very small, but the radial residual stress distribution is always perpendicular to the main load direction. This shows that when the hole spacing is small, the extrusion process demonstrates a negligible effect on the plastic deformation area near the hole edge, but it produces interference and stress superposition in the elastic deformation area away from the hole edge. When the hole spacing is substantially large, the interference behavior is insignificant.

The residual stress distribution curves of the entry plane, mid-thickness, and exit plane where the central hole of the single-hole specimen is located are shown in Figure 18. Figure 18a,b are the circumferential and radial residual stress distributions, respectively. Figure 18a explains that the circumferential residual stress caused by SSCE near the hole wall can be divided into a compressive stress zone and tensile stress zone. The circumferential residual compressive stress is the key to improve the crack initiation life and crack growth life of the connecting hole. Figure 18a shows reverse yield zones at the entry plane, mid-thickness plane, and exit plane at 0–1 mm where the reverse yield and the size of the reverse yield zone are related to the extrusion interference. Since the finite element follows a Cartesian coordinate system, the circumferential residual stress (shown in Figure 18a) is uniformly distributed along the line of symmetry. The results of circumferential compression residual stresses are found in good agreement with the simulation outcomes. These results also match with the effect of the cold expansion of Al7075 holes reported by Chakherlou et al. [38]. From Figure 18b, it can be observed that residual tensile stress occurs at the extrusion end (3–6 mm from the mandrel to the hole). In the SSCE simulation, the gap edges in the split sleeve expand due to the direct contact between the mandrel and the inner wall of the hole, which drags the material through the hole along the axial direction. This process can cause the release of residual stresses and redistribution of residual stresses around the hole. However, the effect of this process is neglected in the finite element simulation [39]. During the SSCE process, the least amount of compressive residual stress occurs in the entry plane, the largest amount of compressive residual stress in the mid-thickness plane, and good compressive residual stress in the exit plane, as shown in Figure 15, Figure 16 and Figure 17. The result has been confirmed by several researchers that, with the increase in thickness, the compressive residual stress at the entrance plane decreases [40,41,42,43].

In either SSCE order, the last extrusion hole is the right hole. The circumferential residual stress distribution curves of the right hole of the specimen for two extrusion sequences with three times the hole diameter as the hole spacing are given in Figure 19 and Figure 20, respectively. The maximum radial residual compressive stress was increased by 31.53% for the three-hole specimen with triple the hole diameter as hole spacing compared to the single-hole residual stress. In contrast, the maximum radial residual compressive stress of the three-hole specimen with four times the hole diameter as the hole spacing was reduced by 2.98% compared to the single-hole specimen. This shows that SSCE with three times the hole pitch can significantly improve the extrusion strengthening effect, while SSCE with four times the hole pitch has no gain in the extrusion strengthening of adjacent holes and can be regarded as single-hole SSCE.

Figure 21 shows the difference between the radial residual stress distribution of the multi-hole SSCE process and the radial residual stress distribution of the single hole. Due to the stress superposition effect, the radial residual stress generated after the middle hole is extruded is transferred to the sides of the hole, so the radial residual compressive stress increases when the right hole is extruded, which improves the SSCE benefit. In addition, the single-hole SSCE process generates tensile residual stresses in the exit plane, while the three-hole SSCE avoids this phenomenon through the stress superposition effect.

It is worth noting that the circumferential compressive residual stress of the three-hole specimen with hole spacing three times the diameter of the hole is not much different from that of the hole spacing of four times, but the radial compressive residual stress is 34.46% higher than that of the three-hole specimen with hole spacing of four times. This shows that the effect of superposition strengthening of residual stress between adjacent holes of three-hole specimens with hole spacing three times the diameter of the hole is more significant. In addition, for the three-hole specimen with hole spacing three times the diameter of the hole, the circumferential compressive residual stress is generated by the SSCE sequence of the middle hole first, then the left hole, and finally that of the right hole is 1.85–7.8% higher than that generated by the SSCE sequence from left to right, and the radial compressive residual stress is increased by 0.54–1.67%. This proves that the SSCE sequence of middle hole first, then left hole, and finally right hole is better than the sequence of left to right hole. Therefore, when performing SSCE on multiple holes in actual working conditions, the SSCE sequence of middle hole, left hole, and right hole can be selected preferentially. However, in order to avoid the superposition of multiple stress fields in the holes adjacent to the left and right holes, the FTI-8101 standard should be considered for the insertion of pins [36]. When the SSCE of these three holes is completed, the pins can be removed and the above operation can be repeated for SSCE of the other holes one by one.

### 3.2. Strain Results of Test Piece Measured by DIC

During the SSCE of the actual split sleeve hole, the von Mises strain in the entry plane is recorded by the DIC equipment to verify the strain change, and the DIC shot results are analyzed by the Vic-2D 7.0image analysis software. Firstly, the graph is captured by the high-speed camera in Vic-2D 7.0. Secondly, the solution location is selected and the hole location is deleted, and the numerical values of noise level and strain resolution are 8 and 15, respectively. Then, the software automatically calculates the solution area as a number of small squares, each with a scattered subset size of 17. The results are obtained by selecting the calculated strain. Figure 22 shows the change in von Mises strain in the entry plane of the single-hole specimen at different stages of extrusion. After the opening of the split, the sleeve faces the radial direction of the specimen, and four directions around the hole are symmetrical. Only the von Mises strain data on the left and lower sides of the hole are recorded. By observing the strain distribution in Figure 22b,c, it can be found that the strain peaks during and after the extrusion process are distributed to the left side adjacent to the opening of the bushing and at the diagonal position (marked by arrows). However, the strain distribution is not obvious in other areas of the hole edge. In addition, the strain fluctuations at the hole edges in Figure 22b,c are transmitted to the edges of the specimen.

Figure 23a shows the von Mises strain on the left and lower sides of the FEM hole, and Figure 23b shows the von Mises strain on the left and lower sides of the DIC hole. Comparing Figure 23a,b, it can be found that the strain variation trend of the single-hole entrance plane is basically identical in the finite element simulation and the DIC shot. The strain gradually decreases from the center of the hole to the outside. However, the finite element-extracted strain curve is very smooth without fluctuations, while the curve extracted by DIC shows fluctuations during and after the SSCE (i.e., extrusion stabilization stage and extrusion rebound stage) at 2–4 mm to the left of the hole. There is a fair amount of noise in the DIC measurements, so the curve produces some corresponding fluctuations. During the SSCE, the mandrel extrusion split sleeve is opened at the opening of the split sleeve, causing a large plastic deformation around the hole and the strain amplitude appears at the edge of the hole. The results are in agreement with those of Amjad for cold expansion of thin-walled parts [44].

Figure 24 shows the strain distributions during SSCE of the middle hole and the right-side hole of a triple-pitch three-hole specimen (including DIC results and finite element results). From Figure 24b,c, it is observed that during the center-hole SSCE, strain peaks at the edge of the hole, and dissipates outward along the hole, while the strain size decreases gradually from the edge of the hole. However, when the middle hole is cold expanded, the strain extends to the center hole along the direction of the minimum cross-sectional area of the hole, indicating that residual strain of the center hole is generated during the SSCE. As shown in Figure 24e,f, the strain interference exists at the lip where the center hole meets the two holes. It is similar to the strain transfer from the edge of the single hole to the edge of the specimen in Figure 22, where the strain at the edge of the hole fluctuates to both sides during the extrusion of the central hole, while the strain at the lip between the middle hole and the right hole is dramatic during the extrusion of the right hole. This indicates that after the completion of extrusion of the center hole, there is still extrusion rebound and strain fluctuation, and the strain generated during the subsequent SSCE process at this time has a superposition effect with the center hole strain in the same direction, so the observed lip strain between the center hole and the right hole changes significantly. Figure 24g–i give the strain results during the extrusion of the right-hand side hole of the finite element (dissected along the *y*-axis in the central hole), where the strain increases continuously during the initial stage of extrusion, begins to decrease during the stabilization stage of extrusion, and remains essentially constant after extrusion. In comparison with the DIC results in Figure 24e,f, the lip strain transfer fluctuations between the central and right hole are evident when the right hole is extruded.

Figure 25 shows the strain results for the three-hole specimen with three times the hole diameter as the hole spacing according to the sequential SSCE from left to right. As can be seen in Figure 25b,c, due to the sequential extrusion from left to right, there is extrusion rebound in the left-side hole during the extrusion of the center hole, which generates a superposition in the lip between the left-side hole and the middle hole with significant strain. Similarly, in Figure 25e,f, the strain in the lip between the center hole and the right hole varies significantly. Figure 25g–i similarly show this phenomenon. Comparing the magnitude of the strain values under two different SSCE sequences in Figure 24 and Figure 25, it is found that the strains taken by DIC all vary in the interval from 0 to 0.1, and the FEM results are all slightly larger than the DIC results. On one hand, this is due to noise interference during the DIC shot. On the other hand, this is because the FEM controls the SSCE process by displacement in order to ensure the continuity of the SSCE process and does not ensure the extrusion under a certain constant load.

Figure 26 shows the results of sequential SSCE with four times the hole pitch. The strain results shown in Figure 26a–f are the same as those of sequential SSCE with three times the hole pitch. A small interference lies between the holes of quadruple-pitch SSCE. When SSCE occurs in the right-side hole, the strain on the middle hole interference is transmitted more at 45 degrees to the left of the split sleeve opening (shown with arrows). From Figure 26g–i, it can be observed that the strain fluctuations spread from the edges of the extruded holes in all directions and cannot be transmitted to the adjacent holes for the large hole spacing, so the effect on the adjacent holes is relatively small. Combined with the results of FEM, the radial residual stress of three-hole specimens with hole spacing four times the hole diameter is lower than that of single holes, which can provide theoretical support for process optimization: in the case of multi-hole SSCE, when the hole spacing is greater than four times the hole diameter, it can be regarded as single-hole SSCE, and the interference effect between holes can be neglected. Comparing the pattern of strain fluctuation at different hole spacings, the results show that the residual strain superposition between consecutive rows of holes is related to the hole spacing. SSCE of multiple holes at hole spacing three times the hole diameter is more effective in strengthening the specimen, while when the hole spacing is greater than or equal to four times the hole diameter, SSCE has no strengthening effect on adjacent holes and can be regarded as single-hole SSCE.

For the three-hole test piece, the test piece is completely symmetrical, therefore, only the von Mises strain at the hole edge of the middle and the right hole is analyzed. The DIC shot results are processed by Vic-2D and the strain variation on the four straight lines $L_{0}$, $L_{1}$, $L_{2}$, and $L_{3}$ are selected from Figure 24a,d, Figure 25a,d and Figure 26a,d. The strain variation trend around each hole of the specimen is analyzed. Figure 27a–c show that the left side of the center hole during the initial SSCE of the middle hole produces the highest strain which is followed by the lower side of the center hole. At the same time, the strains on the left side and the lower end of the right side of the hole change during the SSCE of the center hole. Similarly, Figure 27d–f show the maximum strain on the left side of the right hole followed by the lower side of the right hole, while the strains on the left and lower sides of the center hole also change drastically, indicating that the strain interference dominates in this SSCE sequence. Figure 28a–c show that only the strain on the left side of the hole ($L_{0}$) changes when the center hole is extruded, while no significant change is observed in the strain on the lower side of the hole as well as to the left and lower sides of the right hole. However, for SSCE of the right hole, Figure 28d–f show the maximum strain values on the lower side of the right hole ($L_{1}$) followed by the left side of the right hole ($L_{0}$), and a small fluctuation is noted in the strain of the center hole, indicating that the same strain interference occurs under sequential SSCE. From Figure 29a–c, it can be seen that the results are consistent with the results of the center hole of the three-hole sequential SSCE, and the SSCE of the center hole at hole spacing four times the hole diameter is also significant only for the strain on the left side ($L_{0}$) of the center hole. The strains on the left side ($L_{2}$) and the lower side ($L_{3}$) of the center hole remain unchanged during the SSCE of the right-side hole, which indicates that the hole spacing interferes with the strains, and the interference exists at hole spacing three times the hole diameter, but no such phenomenon is observed at hole spacing four times the hole diameter. This confirms that the effect of SSCE in the middle hole of multi-linked holes is better than that of sequential SSCE, and the reinforcement effect of triple SSCE is better than the quadruple SSCE.

Comparing Figure 27 with Figure 28, it can be seen that under different SSCE sequences, the strain variation interval corresponding to the squeezing sequence of squeezing the middle hole first at three times the hole pitch fluctuates between 0 and 0.025, and the strains on the left and lower side of the hole edge fluctuate when squeezing the center and right holes; for the sequential squeezing at three times the hole pitch, the strain variation corresponds to a larger span, and only the left side of the hole changes significantly when squeezing the center and right holes. This indicates that, on one hand, the strain produces a superposition effect in the direction of hole alignment. For the result of sequential extrusion, the strain on the left side of the hole edge is always large because the left hole of sequential extrusion completes SSCE first, and the lip strain between the center hole and the left hole is significant in the center hole SSCE, while the lip strain between the center hole and the right hole is significant in the right hole SSCE. However, in the order in which the center hole is extruded first, after completing the center hole SSCE. The center hole and right hole SSCE processes are not continuous, so the curves show corresponding changes in strains in both the left hole and the lower hole. On the other hand, it also shows the strain change law, that is, when the front end of the working section of the mandrel is in the hole, the extrusion pressure and strain reach the maximum value at the same time; when the front end of the working section of the mandrel is inside the hole to the rear, that is, the extrusion stabilization stage, the extrusion pressure is basically unchanged and the strain decreases; when the back end of the working section of the mandrel is in the hole and detaches from the hole, i.e., the extrusion rebound phase, the strain is constant and the extrusion pressure decreases.

Comparing Figure 28 and Figure 29, it can be found that the trend of the strain variation on the left side of the hole is the same as that of the single hole when the center hole is extruded at four times the hole pitch, while the strain on the lower side is weak; when the right hole is extruded, the strain on the left side and lower side of the center hole basically does not change, and only the strain on the lower side decreases slightly during the extrusion stabilization stage (red line in Figure 29e), but the overall strain change on the left side and lower side of the right hole is the same as that of the single hole, which confirms that the inter-hole interference effect decreases with the increasing hole pitch during SSCE. Furthermore, combining the three-hole strain profile with Figure 23 shows that the strain when SSCE is carried out in any hole in the three-hole SSCE is always smaller than the single-hole SSCE strain, indicating that there is strain interference between the multi-hole SSCE and the adjacent holes. The strain itself decreases after the extrusion stabilization and is smaller when the neighboring holes interfere.

## 4. Conclusions

The residual stress distribution at the hole edge under the single-hole and three-hole SSCE was studied through finite element simulation, and the effects of the SSCE process on the residual stress around adjacent holes were compared. Combining the strain distribution at the edge of the hole obtained by FEM and DIC and the interference effect between adjacent holes, the results can be summarized as follows:(1)A mandrel extrusion split sleeve may occur along the axial drag during the SSCE for single-hole specimens, and the exit radial plane generates tensile residual stress. However, in the case of multi-hole SSCE, the superposition of stress optimizes this phenomenon.(2)Compared with the single-hole SSCE, the radial residual compressive stress at the hole edge increases significantly during the three-hole SSCE due to the stress superposition effect, and this gain effect is more significant in the order of center hole, then left hole, and right hole. When SSCE is performed for multiple holes in actual working conditions, the SSCE sequence of middle hole, left hole, and right hole can be selected preferentially. However, in order to avoid the interference of the left and right holes with the next SSCE, it is necessary to use inserting pins for the adjacent left and right holes, and when the SSCE of these three holes is completed, the pins are removed and the sequence is repeated to perform SSCE for other holes in sequence.(3)The hole edge strain results of FEM and DIC are in good agreement. The single-hole strain results show that the hole edge strain amplitude is concentrated at the left side of the split sleeve opening and at the diagonal position, and the strain tends to transfer to the specimen edge. The multi-hole strain results further verify that the hole edge strains are transmitted in the direction of hole alignment and produce a superposition effect.(4)When the hole spacing is greater than or equal to four times the hole diameter, the strain distribution under SSCE in the adjacent hole interference is reduced and the corresponding strengthening effect is smaller. Combined with the FEM results, the interference effect between the adjacent holes with spacing four times the hole diameter can be ignored, and in the actual process the inter-hole spacing between multiple holes can be regarded as single-hole SSCE if it is greater than four times the hole diameter.(5)Based on residual compressive stress results of both circumferential and radial components, it appears that the configuration with the smaller distance between the SSCE holes is the most beneficial. However, if the plate is subjected to a tensile plane deformation, for example, by applying a force normal to the line formed by the centers of three holes, tensile stress is created in the material between the holes. Such tensile stress increases as the hole spacing decreases. Thus, these opposite effects must be considered when optimizing the configuration of the holes in the airframe.

This paper still has some limitations. Although it has been verified through finite element simulation and experiments that the multi-hole specimen SSCE can provide guidance for process optimization with hole spacing three times the hole diameter, the case of hole spacing which less than three times the hole diameter has not been explored. Moreover, the paper is focused on aerospace aluminum alloy plate, but it is still unknown whether this process is applicable for other materials. Therefore, the future research direction will be directed to the above two points.

## Figures and Tables

**Figure 1 materials-16-01109-f001:**
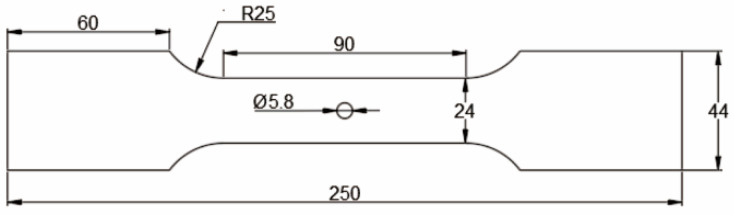
Single-hole aluminum alloy specimen (without cold expansion).

**Figure 2 materials-16-01109-f002:**
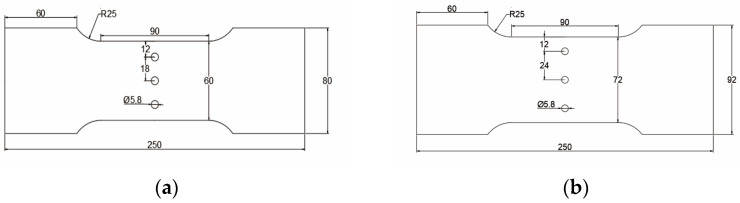
Three-hole aluminum alloy specimen (without cold expansion): (**a**) hole spacing is 3 times the final hole diameter, (**b**) hole spacing is 4 times the final hole diameter.

**Figure 3 materials-16-01109-f003:**
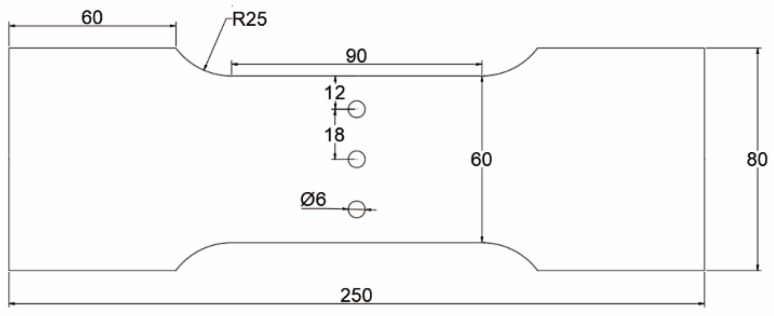
Three-hole cold-expanded aluminum alloy specimen.

**Figure 4 materials-16-01109-f004:**
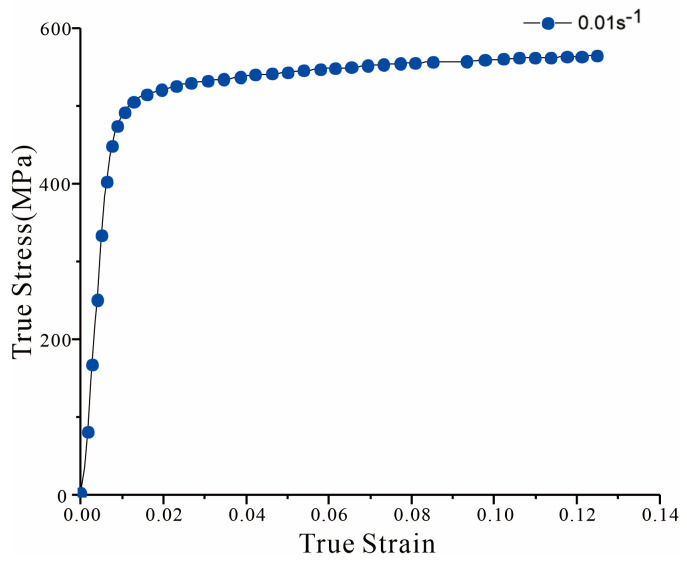
Stress-strain curve of Al7075-T6 at quasi-static strain rate.

**Figure 5 materials-16-01109-f005:**
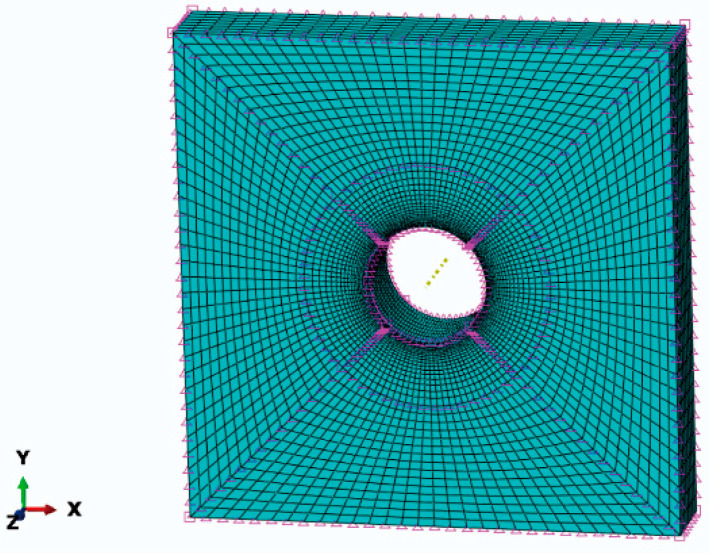
Single-hole specimen hole edge area meshing.

**Figure 6 materials-16-01109-f006:**
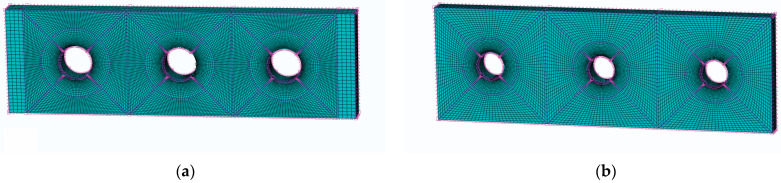
Mesh quality of three-hole specimen hole edge: (**a**) three times the hole diameter for the three-hole specimen; (**b**) four times the hole diameter for the three-hole specimen.

**Figure 7 materials-16-01109-f007:**
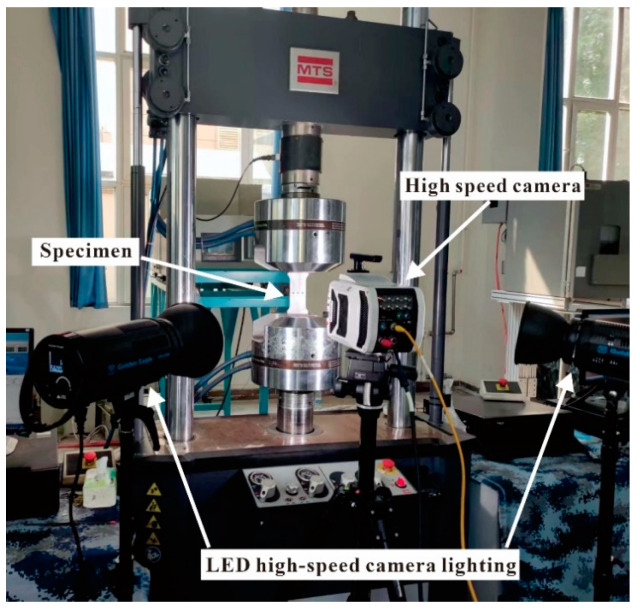
High-speed camera shooting test piece and supporting LED camera lighting equipment.

**Figure 8 materials-16-01109-f008:**
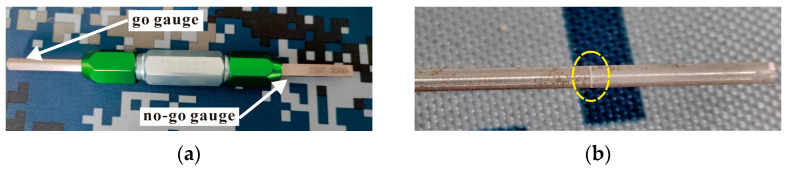
(**a**) FTI standard go/no-go gauge; (**b**) the no-go gauge bayonet.

**Figure 9 materials-16-01109-f009:**
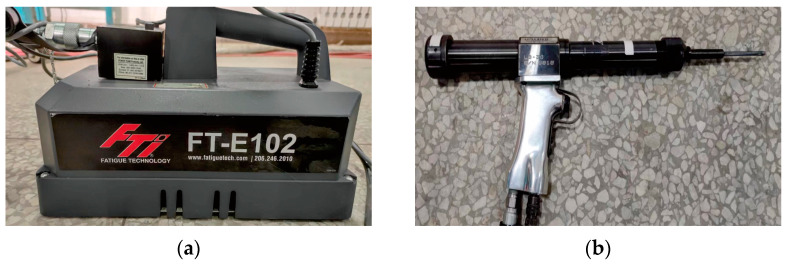
SSCE equipment: (**a**) oil pump; (**b**) pulls gun; (**c**) mandrel; (**d**) split sleeve.

**Figure 10 materials-16-01109-f010:**
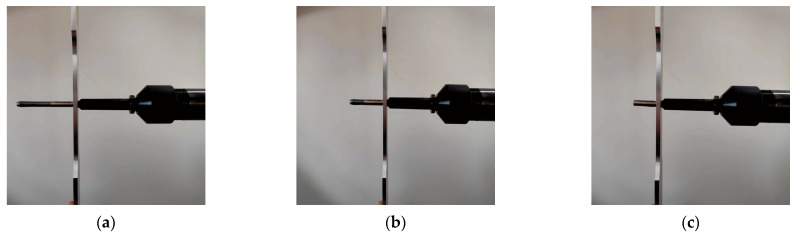
Explanation of the SSCE process: (**a**) the split sleeve and mandrel pass through the hole; (**b**) the pulling gun is started and the core rod is extruded through the hole with the split sleeve; (**c**) SSCE is completed.

**Figure 11 materials-16-01109-f011:**
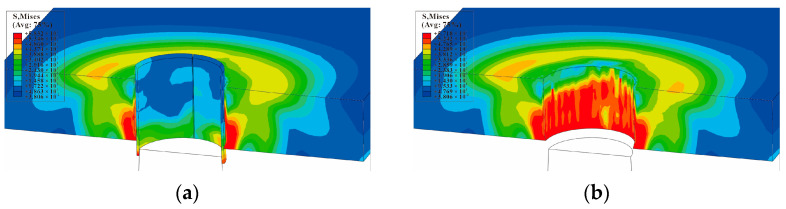
Equivalent von Mises stress at the cross-section of single-hole SSCE: (**a**) equivalent von Mises stress extruded from a single hole without a hidden split sleeve; (**b**) equivalent von Mises stress after extrusion of the single hole with concealed split sleeve (the split sleeve is hidden to show clear stress contours).

**Figure 12 materials-16-01109-f012:**
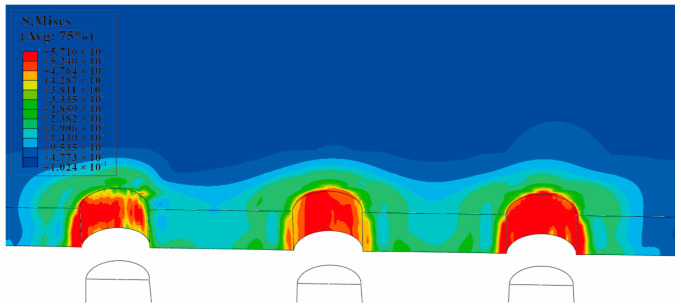
Equivalent von Mises stress of each hole along the *x*-axis of a three-hole cold-expanded specimen with a hole spacing of three times the diameter of the hole.

**Figure 13 materials-16-01109-f013:**
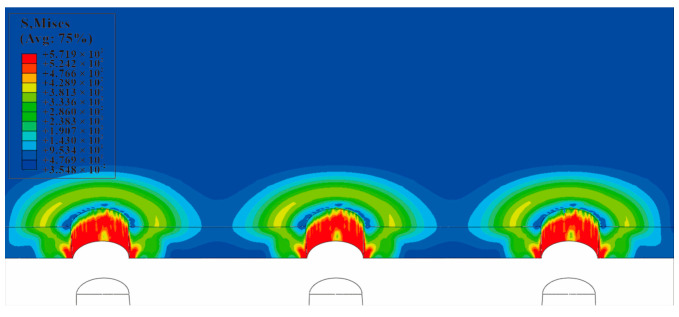
Equivalent von Mises stress of each hole along the *x*-axis of a three-hole cold-expanded specimen with a hole spacing of four times the diameter of the hole.

**Figure 14 materials-16-01109-f014:**
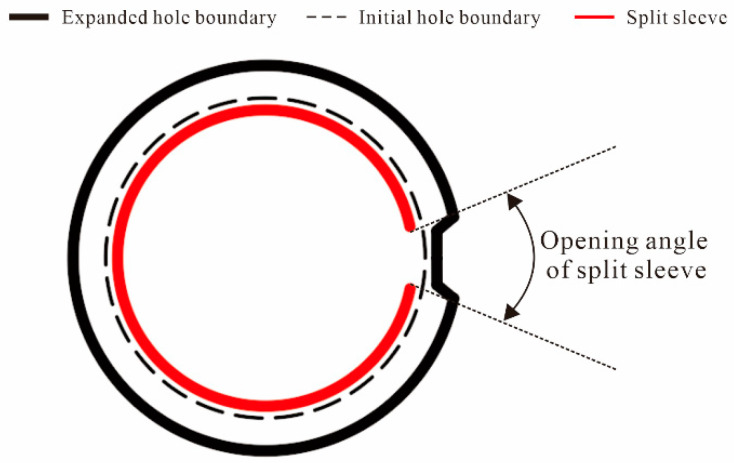
Schematic diagram of the influence of opening on the deformation during the extrusion of split sleeve (the opening angle is drawn at an arbitrary angle).

**Figure 15 materials-16-01109-f015:**
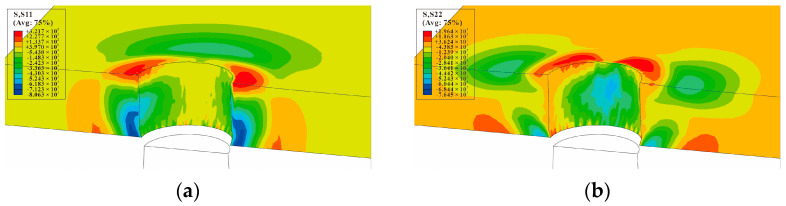
The nephogram of circumferential and radial residual stress around the hole of single-hole SSCE when the mandrel extrudes the middle hole: (**a**) S11/circumferential stress distribution nephogram (MPa); (**b**) S22/radial stress distribution nephogram (MPa).

**Figure 16 materials-16-01109-f016:**
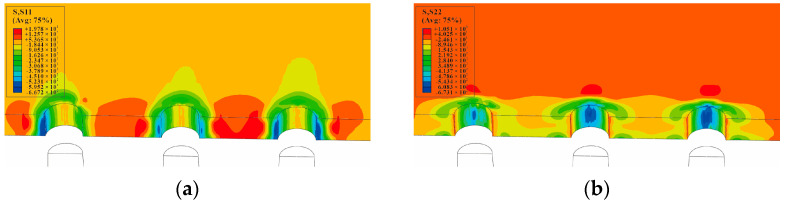
SSCE of the sample with three times the hole diameter as the hole spacing. The circumferential and radial residual stress nephogram at the hole edge after CE: (**a**) S11/circumferential stress distribution nephogram (MPa); (**b**) S22/radial stress distribution nephogram (MPa).

**Figure 17 materials-16-01109-f017:**
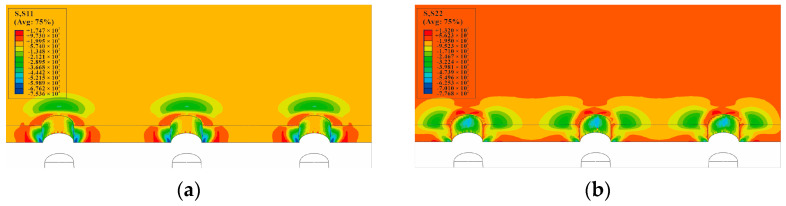
SSCE of the sample with hole spacing four times the hole diameter. The circumferential and radial residual stress nephogram at the hole edge after CE: (**a**) S11/circumferential stress distribution nephogram (MPa); (**b**) S22/radial stress distribution nephogram (MPa).

**Figure 18 materials-16-01109-f018:**
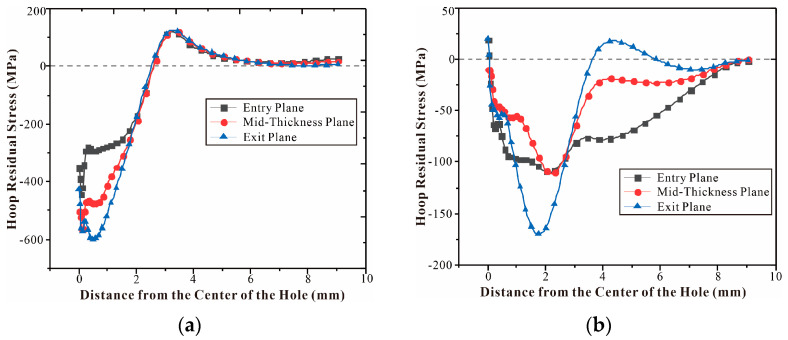
Residual Stress Distribution on the minimum section with the central hole of the single-hole specimen: (**a**) S11/circumferential residual stress distribution; (**b**) S22/Radial residual stress.

**Figure 19 materials-16-01109-f019:**
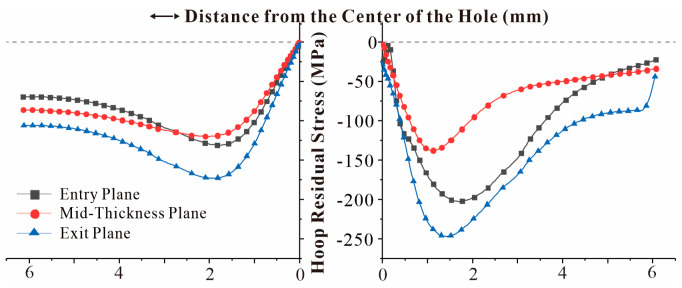
Residual stress distribution in the right hole radial direction for a three-hole specimen with triple aperture spacing (first, the middle hole SSCE, then the left hole, and finally the right hole).

**Figure 20 materials-16-01109-f020:**
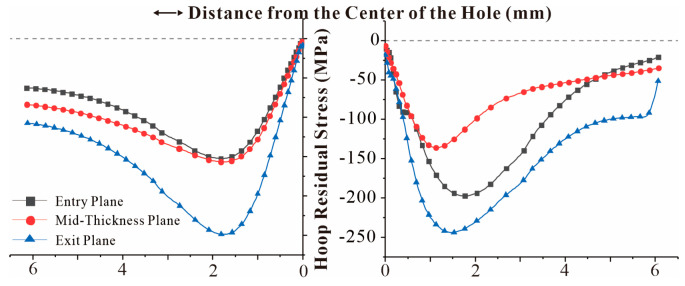
Residual stress distribution in the right hole radial direction for a three-hole specimen with triple aperture spacing (SSCE from left to right).

**Figure 21 materials-16-01109-f021:**
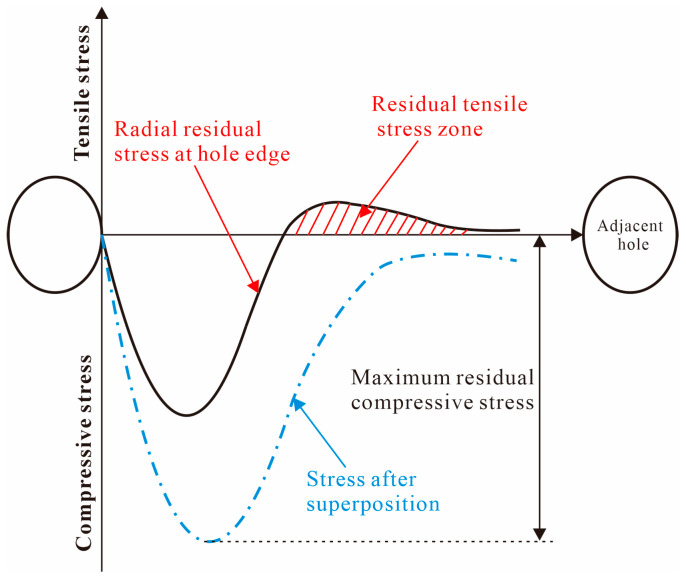
Difference of radial residual stress distribution between porous SSCE and single-hole SSCE.

**Figure 22 materials-16-01109-f022:**
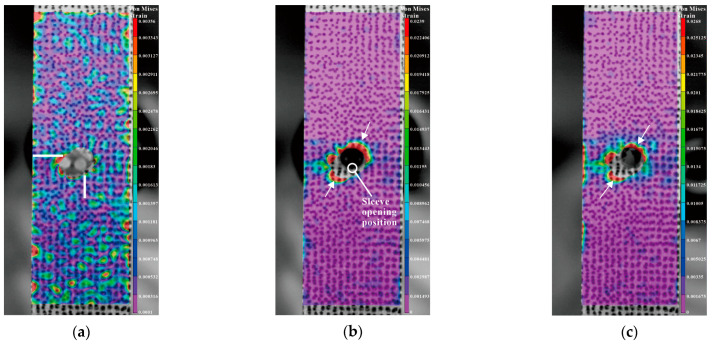
Change in von Mises strain on single-hole specimens during extrusion (Entry Plane): (**a**) before SSCE; (**b**) during SSCE; (**c**) after SSCE.

**Figure 23 materials-16-01109-f023:**
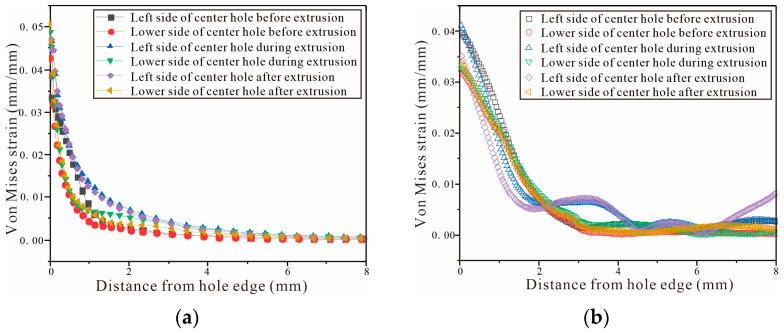
Von Mises strain of the single-hole test piece (entry plane): (**a**) von Mises strain of the left and lower sides of the hole by FEM; (**b**) von Mises strain of the left and lower sides of the hole by DIC.

**Figure 24 materials-16-01109-f024:**
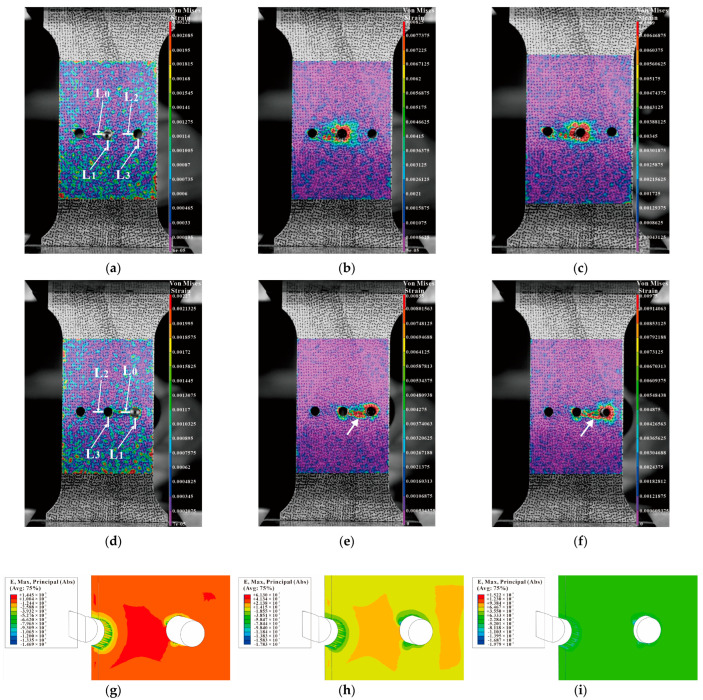
Comparison between DIC strain detection and FEM-predicted strain of three-hole specimen with three times the hole diameter as hole spacing (first middle hole is squeezed, then left hole, and finally right hole): (**a**–**c**) strain distributions before, during, and after SSCE of the middle hole; (**d**–**f**) strain distributions before, during, and after SSCE of the right hole; (**g**–**i**) the right-hole SSCE strain distribution of FEM.

**Figure 25 materials-16-01109-f025:**
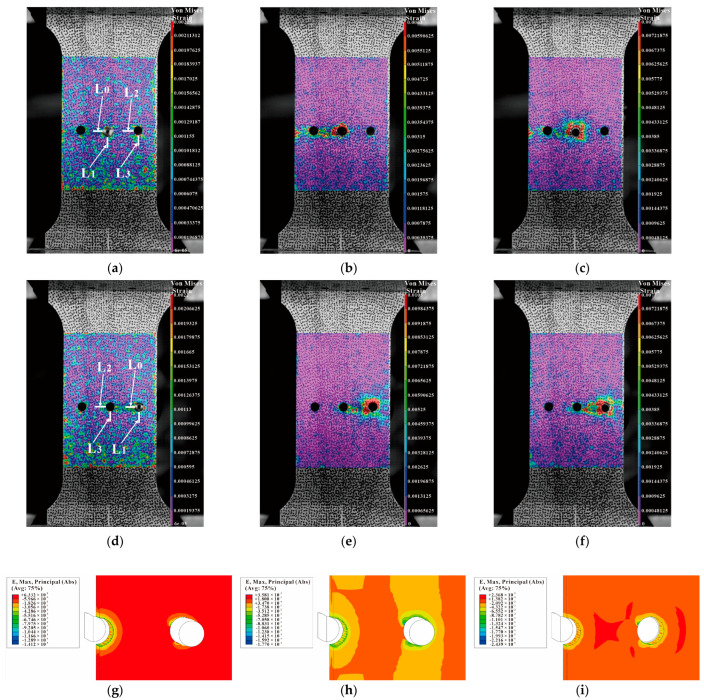
Comparison between DIC strain detection and FEM-predicted strain of three-hole specimen with three times the hole diameter as hole spacing (sequential extrusion): (**a**–**c**) strain distributions before, during, and after SSCE of the middle hole; (**d**–**f**) strain distributions before, during, and after SSCE of the right hole; (**g**–**i**) the right-hole SSCE strain distribution of FEM.

**Figure 26 materials-16-01109-f026:**
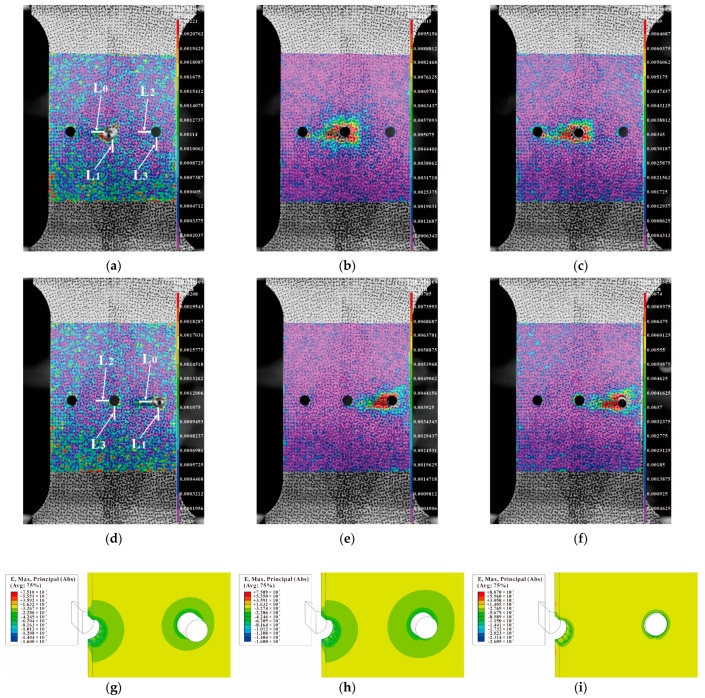
Comparison between DIC strain detection and FEM-predicted strain of three-hole specimen with four times the hole diameter as hole spacing (sequential extrusion): (**a**–**c**) strain distributions before, during, and after SSCE of the middle hole; (**d**–**f**) strain distributions before, during, and after SSCE of the right hole; (**g**–**i**) the right-hole SSCE strain distribution of FEM.

**Figure 27 materials-16-01109-f027:**
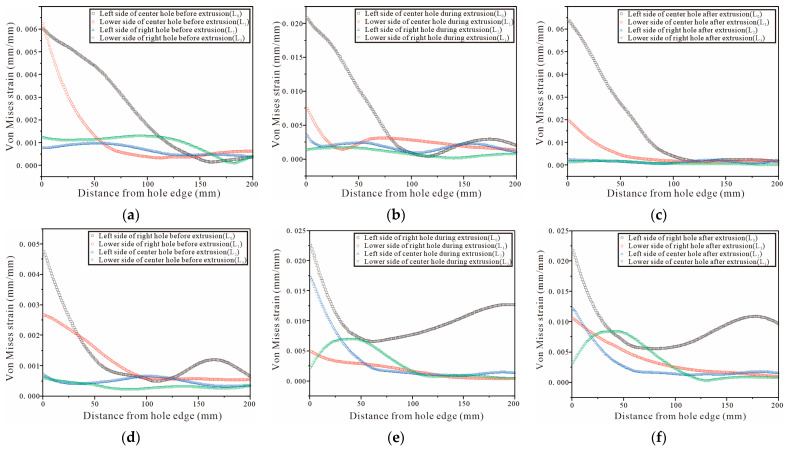
Von Mises strain for the left and lower sides of triple-pitch three-hole specimens (first, the center hole is cold expanded, then the left hole, and finally the right hole): (**a**–**c**) strain distributions before, during, and after SSCE of the middle hole; (**d**–**f**) strain distributions before, during, and after SSCE of the right hole.

**Figure 28 materials-16-01109-f028:**
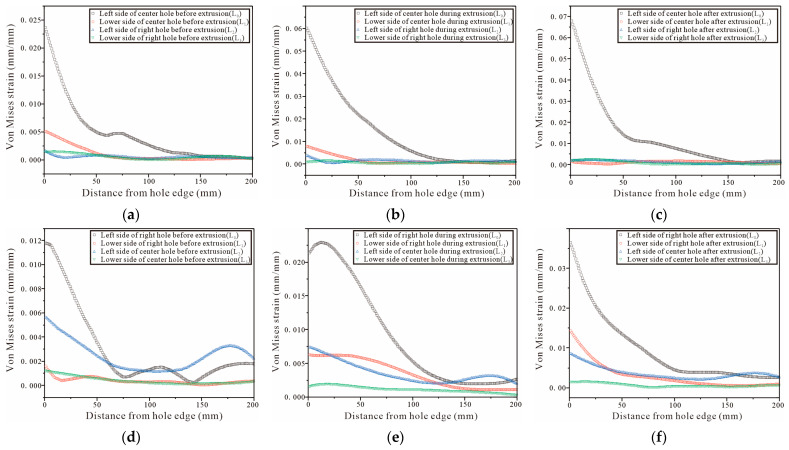
Von Mises strain results for the left and lower sides of the triple-pitch three-hole specimens (sequential SSCE): (**a**–**c**) strain distributions before, during, and after SSCE of the middle hole; (**d**–**f**) strain distributions before, during, and after SSCE of the right hole.

**Figure 29 materials-16-01109-f029:**
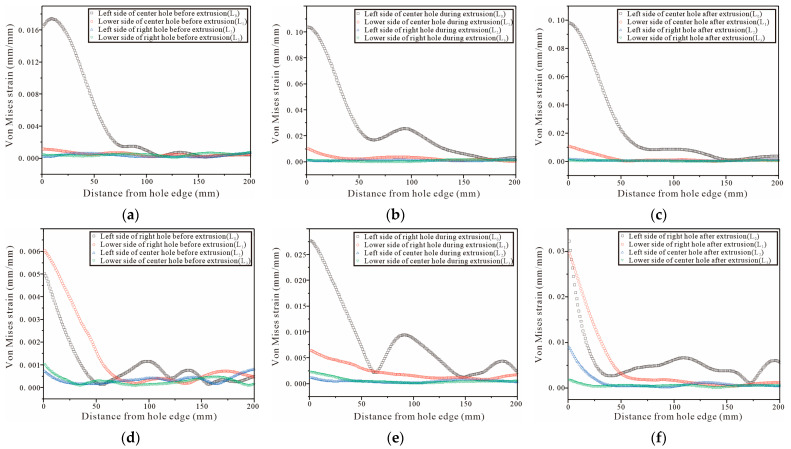
Von Mises strain results for the left and lower sides of the quadruple-pitch three-hole specimens (sequential SSCE): (**a**–**c**) strain distributions before, during, and after SSCE of the middle hole; (**d**–**f**) strain distributions before, during, and after SSCE of the right hole.

**Table 1 materials-16-01109-t001:** Chemical composition of Al7075-T6.

Element	Si	Fe	Cu	Mn	Mg	Cr	Zn	Ti	Al
Quality Score/%	0.03	0.22	1.55	0.09	2.38	0.19	5.4	0.02	remainder

**Table 2 materials-16-01109-t002:** 7075-T6 Plate Thickness and Mechanical Properties.

Material	Thickness (mm)	Tensile Strength (MPa)	Yield Strength (MPa)	Elongation (%)
7075-T6	4	556–566	506–509	11.5–12.5

## Data Availability

All the data are available within the manuscript.

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
