# Peer review of "Finite Element Analysis of Split Sleeve Cold Expansion Process on Multiple Hole Aluminum Alloy"

_materials, 2023, doi:10.3390/ma16031109_

Round 1

Reviewer 1 Report

This manuscript is devoted to studying the finite element analysis of split sleeve cold expansion process on multiple-hole aluminum alloy. The manuscript is well-written and well organized and includes some merits and may be considered for publication. I am raising some important point which requires attention:

1. Add some recent literature on fatigue loading and fatigue fracture.

2. Provide the manufacturer/supplier details of Al7075-T6.

3.  Provide suitable references: "Two different expression methods are used..." 

4. " hole diameter are shown in Fig. 11 and 18, respectively..." Please correct the Figure number. (line 207)

5.  It is better to add some future directions and outlook of the current work.

Author Response

Dear Editors and reviewers,

On behalf of my co-authors, we thank you very much for giving us an opportunity to revise our manuscript, we appreciate editors and reviewers very much for their positive and constructive comments and suggestions on our manuscript entitled “Finite Element Analysis of Split Sleeve Cold Expansion Process on Multiple Hole Aluminum Alloy”.

We have studied reviewer’s comments carefully and have made revision which marked in red in the paper. We have tried our best to revise our manuscript according to the comments. 

Reviewer 2 Report

This paper describes a Finite Element analysis of a cold expansion process of a multihole specimen. The effect of having multiple holes, as well as the spacing of the holes, is studied. Also DIC measurements are performed on the same specimens. General insights on the magnitude of introduced residual stresses around the holes are given, assisting in optimizing this process.

Although the results are described quite extensively, some details on the followed approach are missing:

-       *  Section 2.3: Details on setting up the FE model and execution of the analyses is rather limited, which makes it non-reproducible. Could the authors add more details on how the contact between the bodies has been modelled, how the movement of the mandrel has been simulated (time steps, implicit / explicit integration scheme, motivation of these choices, etc.) ?

-       *  Also details on the used constitutive / material model are missing: how is plasticity modeled ? What material constants (E-modulus, strain hardening, ...) are used, and how have these been determined ? Are they representative for the studied material ?

-        *  Fig. 27 is the only figure comparing the FE and DIC results. Please include a thorough discussion on the observed differences (shape of curves, absolute values, etc.). This is required to validate the FE results, i.e. prove that the simulations correctly represent what is observed in an experiment ! Moreover, add FE result plots that can be compared with the DIC results in Fig. 28 / 29. This allows the reader to assess the quality of the FE results.

-           

 Further, the purpose and added value of the work are not very clear:

-        *  In the introduction it is mentioned that “it is urgent to carry out multi-row hole research” and “the SSCE process of three-hole has not been studied in depth”. However, it is not described why the authors think this would be interesting. Are very different results expected than from previous research (on single / double holes) ?

-        It is not clearly explained why FE analyses must be made to obtain the presented insights. It seems that just performing the DIC analyses would also give these insights. So why add the FE analyses ?

-        * The results summarized in the conclusion (section 4) are every specific for this case (material, geometry). It has not been discussed whether these results could be generalized to other geometries, materials, loads, etc. ? Therefore, the scientific contribution is rather limited, since an engineering problem (for a specific case) has been solved.

Some more detailed comments:

-          In the text, the use of articles (the, a(n)) is rather weak: many are missing, or used in the wrong manner.

-          The title of section 2.1 is “Microstructure of the as-received coating”. It is unclear why a coating is mentioned, there seems to be no coating

-          The title of section 2.2 is “Define plasticity in ABAQUS”. This is inappropriate, as the section is not about plasticity, rather on strain definitions.

-          Various sentences are badly formulated, e.g.

o   P2, line 93: … the investigation of the residual stress distribution law at the edge of single-hole and three-hole cold expanded holes (specimens?), the effect of different hole spacing, and expansion sequences on the stress interference between adjacent cold-expanded (words missing ?).

o   P4, line 117: the size of the extrusion amount of the split sleeve depends on the ratio of the material to be extruded (what is this?), the size of the aperture (what is this?)

o   P5, line 136: Considering the incompressibility of plastic deformation and assuming that the elastic deformation is also incompressible, …: if something deforms, it can not be incompressible !

o   Line 139: current area A : why current ?

o   Line 160: effort, the seeds are distributed in an offset manner with an eccentricity of 15, and 30 units are applied on each slope : not clear what has been done here …

-          Page 3: both tables are numbered Table 1

-          Figure 5: Why show both meshes ? Are meshes different ?

-          Line 176: reference to Fig 11(a) seems to be wrong;

-          Line 207: reference to Fig 11 and 18 seems to be wrong

-          The plots in Fig 10-12 and 14 – 16 would benefit from a zoom-in to the hole regions. Now the important region is very small, and hard to analyse

-          The resolution (quality of graphic) of Fig 18-25 is rather poor

-          The authors tend to give a lot of results in many figures (both for FE and DIC). That makes it hard to follow the reasoning. It is advised to only show typical plots, and only the ones that are used to discuss remarkable observations. All other figures can then be removed.

-          Section 3.2, line 367 – 370: the description of positions with angles is very unclear. Consider to add arrows or letters in the figure at certain positions, and refer to those in the text.

-          The DIC photos in Fig 26 are way too small, they can not be properly studied in this way. For all other DIC photo’s: the legend is way too small, and can not be read !

-          Line 383 and 398: statements like “the strain distribution matches the single-hole SSCE strain results” and “The strain results are shown in Figs. 30(a) to 30(f) are the same as those …” are too generic. Explain why / how results match: in terms of shape, value, …? In the conclusion (section 4) it is also stated “3D finite element simulations are in good agreement with the DIC full-field strain results”, but that has not convincingly been demonstrated in the paper !

-          Section 4 is called Discussion, but contains the conclusions (and a Conclusion section is missing)

Author Response

(The authors gave the same response as above.)

Reviewer 3 Report

The manuscript investigates the importance of cold expansion and split sleeve cold expansion methods for creating compressive residual stresses in single and multiple holes present in aluminum 7075-T651 plate. The topic is relevant in supporting design of aircraft structures. The manuscript includes both numerical and experimental studies.

·        Section 2.2 titled “Define Plasticity in Abaqus” is a description of the general definition of the engineering strain/ engineering stress and not necessarily a plasticity law. Can the authors explain more clearly what is the material model used in Abaqus? Is it an elastic model or elastic plastic? In case of elastic isotropic material, the second constant (for example Poisson Ratio or shear modulus) needs to be specified on addition to Young modulus. If plasticity is considered, is the model perfectly plastic or there is hardening. What is the difference in material properties for the plate versus split sleeve?

·        Eq 9 is labeled as 5.

·        The finite element results were reported in terms of von Mises, circumferential and radial stress (Figures 18-25). On the other hand, all measurement results are presented in terms of von Mises Strain (Figures 26-33). I strongly recommend direct comparison between model predictions and experiments, maybe include the FE total strain in some of the plots in Figures 31-33.

·        DIC measurements. As it can be observed in Figures 28-30 there is a fair amount of noise the DIC measurements (and it is typical for macroscopic DIC techniques). But the strain curves plotted in Figures 31-33 seem fairly smooth. Can the authors explain the smoothing procedure made to estimate the DIC strains – initial resolution and the area on which the averaged was made?

·        Conclusion 4. Lines 474-479. Based on residual compressive stress of both circumferential and radial components, it appears the configuration with the smaller distance between the SSCE holes is the most beneficial. However, if the plate is subjected to a tensile plane deformation for example by applying a force normal to the line formed by the centers of three holes, tensile stress is created in the material between the holes. Such tensile stress increase as the space between holes decreases. Thus, for an optimal configuration both of these opposite effects must be considered when designing the holes in the airframe.

Author Response

(The authors gave the same response as above.)

Reviewer 4 Report

First of all, it is necessary for authors to present all parameters (temperature & time of stages) of applied T6 treatment used for current alloy.

May be, the formulas (1-10) need in some references.

The FTI-8101 standard & SSCE equipment description must be accompanied with corresponding references to the FTI official site & other sources.

Used DIC method needs in more detailed description then it done in lines 169-170 & 360-362.

What is the “Vic-2D”? Which software was used to images analysis?

The discussion is weak. The detailed comparison of finit elements simulations with the DIC results must be presented & well described. The novelty of DIC studies results must be highlighted.

The figures18-26 & 28-33 are presented in insufficient magnification. May be, the majority of presented figures is better to transfer to the article supplement materials?

Author Response

(The authors gave the same response as above.)

Round 2

Reviewer 4 Report

I think that the paper is ready to publication now.